# Testosterone replacement in young male cancer survivors: A 6-month double-blind randomised placebo-controlled trial

Jennifer S. Walsh[1], Helen Marshall[2], Isabelle L. Smith[2], Diana M. Greenfield[3], Jayne Swain[2], Emma Best[2], James Ashton[4‡], Julia M. Brown[2], Robert Huddart[5], Robert E. Coleman[1], John A. Snowden[6], Richard J. Ross[1]*

1 Department of Oncology and Metabolism, University of Sheffield, Sheffield, United Kingdom, 2 Clinical Trials Research Unit, University of Leeds, Leeds, United Kingdom, 3 Specialised Cancer Services, Sheffield Teaching Hospitals NHS Foundation Trust, Sheffield, United Kingdom, 4 TRYMS Trial Management Group, Sheffield, United Kingdom, 5 Institute for Cancer Research, London, United Kingdom, 6 Department of Haematology, Sheffield Teaching Hospitals NHS Foundation Trust, Sheffield, United Kingdom

‡ This author is a lay representative of the TRYMS study.
* r.j.ross@sheffield.ac.uk

**Data Availability Statement:** All relevant data are within the manuscript and its Supporting Information files.

## Abstract

### Background

Young male cancer survivors have lower testosterone levels, higher fat mass, and worse quality of life (QoL) than age-matched healthy controls. Low testosterone in cancer survivors can be due to orchidectomy or effects of chemotherapy and radiotherapy. We have undertaken a double-blind, placebo-controlled, 6-month trial of testosterone replacement in young male cancer survivors with borderline low testosterone (7–12 nmol/l).

### Methods and findings

This was a multicentre United Kingdom study conducted in secondary care hospital outpatients. Male survivors of testicular cancer, lymphoma, and leukaemia aged 25–50 years with morning total serum testosterone 7–12 nmol/l were recruited. A total of 136 men were randomised between July 2012 and February 2015 (42.6% aged 25–37 years, 57.4% 38–50 years, 88% testicular cancer, 10% lymphoma, matched for body mass index [BMI]). Participants were randomised 1:1 to receive testosterone (Tostran 2% gel) or placebo for 26 weeks. A dose titration was performed after 2 weeks. The coprimary end points were trunk fat mass and SF36 Physical Functioning score (SF36-PF) at 26 weeks by intention to treat. At 26 weeks, testosterone treatment compared with placebo was associated with decreased trunk fat mass (−0.9 kg, 95% CI −1.6 to −0.3, $p = 0.0073$), decreased whole-body fat mass (−1.8 kg, 95% CI −2.9 to −0.7, $p = 0.0016$), and increased lean body mass (1.5 kg, 95% CI 0.9–2.1, $p < 0.001$). Decrease in fat mass was greatest in those with a high truncal fat mass at baseline. There was no treatment effect on SF36-PF or any other QoL scores. Testosterone treatment was well tolerated. The limitations of our study were as follows: a relatively short duration of treatment, only three cancer groups included, and no hard end point data such as cardiovascular events.

**Funding:** This study was funded by Cancer Research UK. The Grant Number was R/126349; the PI was RJR. URL: https://www.cancerresearchuk.org. The funders had no role in study design, data collection and analysis, decision to publish, or preparation of the manuscript

**Competing interests:** I have read the journal's policy and the authors of this manuscript have the following competing interests: JSW receives speaker's honoraria from Eli Lilly and Sandoz, grant funding from Alexion and Immunodiagnostic Systems, donations of drug from Eli Lilly, Prostrakan (Kyowa Kirin) and Consilient for clinical studies, donations of assay kits from Biomedica, and consulting fees from Shire, Mereo Biopharma, Kyowa Kirin, UCB Pharma and PharmaCosmos. JMB received CRUK research grant funding for this project. JAS is Chair of the NHS England Specialised Commissioning Clinical Reference Group for Blood and Marrow Transplantation. REC has received consulting and speaker fees from Amgen, Astellas, Eisai, Genomic Health, Inbiomotion and Scancell; he is a patent holder for a biomarker developed by Inbiomotion; he is a former employee of prIME Oncology. RH is a member of Partnership in Cancer Centre London, Wimbledon. RJR is a Director of Diurnal Plc and owns stock. No other authors have competing interests.

**Abbreviations:** BMI, body mass index; CONSORT, Consolidated Standards of Reporting Trials; CTCAE, Common Terminology Criteria for Adverse Events; DISF-SRII, Derogatis Interview for Sexual Functioning–Self Report II; DXA, dual-energy X-ray absorptiometry; FACIT, Functional Assessment of Chronic Illness Therapy; ITT, intention-to-treat; LH, luteinising hormone; OR, odds ratio; RSE, Rosenberg Self-Esteem; SF36-PF, SF36 Physical Functioning score; QoL, quality of life; SST, serum-separating tube; TRYMS, Testosterone Replacement in Young Male Cancer Survivors.

## Conclusions

In young male cancer survivors with low-normal morning total serum testosterone, replacement with testosterone is associated with an improvement in body composition.

## Trial registration

ISRCTN: 70274195, EudraCT: 2011-000677-31.

## Author summary

### Why was this study done?

- Young male cancer survivors have lower testosterone levels than the healthy population and frequently close to or just below the lower limit of the reference range.

- A common question of young male cancer survivors to their physician is, would I benefit from testosterone treatment?

### What did the researchers do and find?

- We invited young male cancer survivors aged 25–50 years with borderline low testosterone levels to participate in a placebo-controlled double-blind trial of testosterone replacement therapy.

- Young male cancer survivors who participated in the trial received either a placebo (inactive) gel or active testosterone gel, which they applied each day to their skin for 6 months. Neither the participant nor the physician conducting the study knew whether they were receiving placebo or active testosterone.

- At the beginning and end of the trial, we measured the participant's body composition and quality of life using questionnaires.

- At 6 months, the young male cancer survivors treated with testosterone had a decrease in their fat mass, on average, of 1.8 kg and an increase in lean mass of 1.5 kg; however, the quality-of-life questionnaires did not show any difference between those treated with placebo or testosterone.

### What do these findings mean?

- A young male cancer survivor with a borderline low morning testosterone level may benefit from testosterone replacement, with an improvement in body composition with loss of fat mass and increase in muscle mass.

## Introduction

The number of young cancers survivors is ever increasing, and by the year 2020, it is estimated that there will be half a million survivors of childhood cancer residing in the United States [1]. A consequence of this is the emergence of long-term sequelae, late effects, the most common of which are endocrine disorders that affect up to 50% of adult childhood cancer survivors [2, 3]. Endocrine sequelae of cancer therapy include functional alterations in the hypothalamic–pituitary–gonadal axis [4, 5], and over 50% of male cancer survivors have laboratory evidence of impaired gonadal function and are worried about the risk of infertility [6]. Cross-sectional data show that 27% of young male cancer survivors 25–45 years old have a serum testosterone concentration below the 10th centile of matched controls [7], associated with increased trunk fat mass and worse quality of life (QoL) than matched controls [7, 8].

The Endocrine Society guidelines recommend that the diagnosis of testosterone deficiency is only made in men with consistent symptoms and signs and a clearly low serum testosterone [9]. For most symptoms, the serum testosterone threshold for likelihood of symptoms was around the lower limit of the normal range for young men—i.e., approximately 10 nmol/l. The measurement of luteinising hormone (LH) may differentiate primary from secondary hypogonadism, but it may be high with normal testosterone levels and is not recommended for determining treatment.

Diagnosing testosterone deficiency on clinical grounds in a cancer survivor is complex because they may have other causes for the signs and symptoms of testosterone deficiency, which include increased body fat, reduced muscle bulk, and impaired QoL. Male cancer survivors with circulating testosterone levels that are clearly low (<7 nmol/l; below the normal range in most laboratories) will usually be prescribed testosterone therapy. There is, however, no evidence to guide treatment decisions in male cancer survivors with borderline low testosterone levels 7–12 nmol/l.

Most clinical trials of testosterone treatment have been in older men. In symptomatic men age 65 years or older, raising testosterone concentrations from borderline low to the midnormal range for men age 19–40 years had moderate benefit for sexual function and some benefit for mood and depressive symptoms but no benefit for vitality or walking distance [10]. The smaller number of studies in younger men show that testosterone treatment reduces fat mass and increases lean body mass in healthy young men and improves metabolic profile, sexual functioning, and mood in young hypogonadal men with no history of cancer [11–13]. However, in cancer survivors, there may be other contributory causes to adverse body composition, metabolic profile, and poorer QoL, so it cannot be assumed that these problems are all caused by the lower testosterone or that testosterone treatment will have the same effect as in men with other causes of hypogonadism. There has only been one small study of testosterone treatment in male cancer survivors [14]. Sixteen men were included in the treatment arm, and patients were selected based on high LH rather than low testosterone level. In this study, there was no change in body composition after 12 months of treatment.

We hypothesised that the lower testosterone in male cancer survivors may contribute to the high fat mass and poor QoL found in young male cancer survivors. The Testosterone Replacement in Young Male Cancer Survivors (TRYMS) study was an academically led, charity-funded (Cancer Research UK), study to examine the effect of testosterone replacement, given as directed by the testosterone label, on body composition, QoL, and metabolic profile in male cancer survivors with borderline low serum testosterone levels.

## Methods

### Study design

TRYMS (ISRCTN 70274195, EudraCT 2011-000677-31) was a prospective, multicentre, randomised, double-blind, placebo-controlled, parallel-group, superiority trial. Participants were recruited from 10 secondary care centres in the UK. The study was approved by the Derby UK Research Ethics Committee (11/EM/0164), an independent Health Research Authority committee. All participants gave written informed consent, and the study was conducted according to the Declaration of Helsinki. The study was conducted according to the protocol (S1 Text) and statistical analysis plan.

### Participants

Participants were men aged 25–50 years who were at least 12 months from the end of curative treatment for either testicular cancer, lymphoma, or leukaemia and had a baseline morning serum total testosterone between 7 and 12 nmol/l. Exclusion criteria were as follows: body mass index (BMI) above 35 kg/m$^2$, testosterone treatment within the last 12 months, oral or IV glucocorticoid treatment, active graft versus host disease, type 1 diabetes mellitus, disease or current medication known to have significant effects on body fat mass or distribution, hormone-dependent cancer, heart failure, obstructive sleep apnoea, or any other contraindication to testosterone treatment. Participants were approached by their usual care teams in oncology or late effects clinics in secondary care hospitals in Sheffield, London, Cambridge, Glasgow, Nottingham, Leeds, Manchester, Southampton, and Cardiff.

### Randomisation and masking

Patients were randomised centrally on a 1:1 basis to receive either Tostran 2% (testosterone) gel or placebo gel (with the same packaging and gel appearance). Randomisation was stratified to ensure balance between groups by using minimisation with a random element and the following minimisation factors: serum testosterone eligibility measurement 7.0–9.9, 10.0–12.0 nmol/l; type of previous cancer (testicular cancer, lymphoma, leukaemia); BMI (<25, 25–29.9, 30–35 kg/m$^2$); age (divided at middle of study age range: 25–37, 38–50 years); time from end of curative cancer treatment (12–30, 31–60, 61+ months); and randomising site. An automated telephone randomisation line was used; minimisation factors were entered, and a randomisation number was issued. Participants and all members of local study teams were blind to treatment allocation. Chief Investigators and all other clinical collaborators were also blind to allocation until database lock and primary analyses were complete.

### Intervention

Testosterone replacement or placebo was prescribed, with the licenced recommended starting dose of six pumps, equivalent to 60 mg of Tostran 2%. The gel was administered in the morning to skin of the legs or abdomen. Participants were instructed to apply the gel over an area about the size of both their hands and to wait for the gel to be touch dry before dressing. Serum testosterone was measured in the central laboratory after 2 weeks. A dose titration was performed to bring serum testosterone into the upper half of the reference range, following an algorithm (Table 1), and checked by an independent unblinded doctor if serum testosterone was outside the range from 5 to 40 nmol/l. Participants allocated to placebo also received a titration instruction to maintain blinding. Treatment adherence was assessed by participant-reported doses missed and weighing of returned Tostran canisters.

**Table 1. Dose titration.**

| Testosterone gel N = 68 | | | Placebo N = 68 |
| --- | --- | --- | --- |
| Serum testosterone | Amended dose | N | N |
| <11 nmol/l | 80 mg/day | 5 | 4 |
| 11–14.9 nmol/l | 70 mg/day | 9 | 7 |
| 15–34.9 nmol/l | 60 mg/day (no change) | 36 | 44 |
| 35–40 nmol/l | 40 mg/day | 13 | 9 |
| >40 nmol/l | 40–20 mg/day | 4 | 1 |
| Titration not done | | 1 | 3 |

Algorithm-based adjustment in testosterone gel arm and numbers of equivalent dummy titrations in placebo arm.

## End points

The trial was designed for two coprimary end points: change in trunk fat mass (kg) by dual-energy X-ray absorptiometry (DXA) and SF36-PF at 26 weeks. SF36-PF was selected from the available QoL tools because it was clinically significantly different between cancer survivors and controls in the previous observational study [8].

The secondary end points were whole-body fat mass and lean mass; fasting insulin:glucose ratio; fasting lipids and bone density at 26 weeks; and BMI, waist circumference, and other QoL scores (from the SF36, Functional Assessment of Chronic Illness Therapy [FACIT] Fatigue, Derogatis Interview for Sexual Functioning–Self Report II [DISF-SRII] sexual functioning, and Rosenberg Self-Esteem [RSE] questionnaires) at 13 and 26 weeks [15–18].

## Procedures

End point assessments and safety bloods were obtained at baseline, 13 weeks, and 26 weeks. Whole-body DXA was only obtained at baseline and week 26. In addition to the baseline, 13-week, and 26-week visits, phone calls to collect adverse events, adherence, and concomitant medications were made at 6 weeks, 19 weeks, and 30 days after treatment completion.

Safety bloods were haemoglobin, haematocrit, and PSA. Adverse events were recorded from enrolment to final phone call 30 days after the end of treatment.

## Anthropometry

Height and weight were measured with electronic scales and a stadiometer to 0.1 cm and 0.1 kg. BMI was calculated as weight (kg)/height$^2$ (m). Waist circumference was measured in centimetres measured at the midpoint between costal margin and iliac crest three times, and the mean value was used.

## DXA

Body composition was assessed by whole-body DXA. Lunar and Hologic machines were used by the study centres. These were all standard-model commercially available DXA machines with the capability for whole-body scanning to assess fat mass, fat-free mass, and bone mass. The whole-body scan is divided into regions by automated analysis to determine regional values for the trunk and limbs. Quality assurance (QA) and image acquisition and analysis were done according to a study-specific manual, with separate instructions for Hologic and Lunar machines, derived from the manufacturers' standard protocols and written by a bone densitometry expert with experience of DXA use in multicentre trials. To determine the precision of each machine, a standard spine phantom was scanned 25 times before the first participant

was measured. QA was done at each site by scanning a standard spine phantom on each day that a participant was measured. Sites provided monthly reports of their QA data and scanner servicing and maintenance. All DXA images and scanner QA reports were reviewed centrally by one of the authors with experience of DXA use in clinical practice and research (JSW) and the bone densitometry expert who wrote the study-specific manual.

## Laboratory measurements

Blood samples were obtained after fasting from midnight to between 8 AM and 10 AM (+/− 1 hour).

Testosterone measurements for eligibility and bloods for safety and secondary end points (full blood count, urea and electrolytes, liver function test, calcium, lipid profile, glucose and insulin) were measured in in each recruiting hospital's NHS laboratories according to their local standard procedures.

Serum testosterone for the titration at 2 weeks was measured in a single central laboratory (Sheffield Teaching Hospitals). Measurements of testosterone and LH from baseline and 26 weeks were also made in the central laboratory in a batched analysis at the end of the study. Serum for these measurements was obtained in serum-separating tubes (SSTs), left to clot for 30 minutes, centrifuged at 4˚C and 3,000 rpm for 10 minutes, and then stored in 1-ml aliquots frozen at −80˚C. Both of these measurements were made by immunoassay using a Roche Cobas 8000, e602 module. CVs were as follows: testosterone $\leq$ 6.4%, LH $\leq$ 4.2%.

## QoL

QoL was assessed with the SF36 [15], FACIT Fatigue [16], DISF-SRII sexual functioning [17], and RSE questionnaires [18]. Participants completed the questionnaires on paper in a private room without study staff present. They were given written instructions on how to complete the questionnaires, with assurances of confidentiality, and told they could miss any questions that they were not able or willing to answer. SF36 is scored to 100 (higher score indicates better QoL), physical functioning score is scaled to 100, and the physical component summary and mental component summary are set to a population mean of 50; FACIT is scored to 52 (higher score indicates less fatigue); DISF-SRII is scored to 100 (100 indicates better sexual function); RSE is scored to 30 (higher score indicates higher self-esteem).

## Statistics

The sample size calculations were based on a two-sample $t$ test. The two primary end points were deemed unrelated, so alpha was not split between the end points. Although there was no evidence to indicate that the treatment effect would differ depending on baseline serum testosterone levels, the trial was originally powered to recruit and analyse in two strata defined by eligibility serum testosterone 7.0–9.9 and 10.0–12.0 nmol/l to test for a treatment effect in each stratum. For trunk fat mass, an effect size of 1.7 kg was determined to be clinically meaningful based on clinical consensus. With a common standard deviation of 3 kg for the change in trunk fat mass [19], 80% power, and a 5% (two-sided) significance level, 50 participants were required per group (total 200 participants). For the SF36-PF, a moderate difference (standardised effect size of 0.5–0.8) was deemed to be clinically meaningful. Between 52 and 128 participants per stratum were required to detect this degree of superiority using a two-sample $t$ test of equal means with 80% power and a 5% (two-sided) significance level (total 104–256 participants). Because of the large range in sample size requirements for SF36-PF, the decision was made to base the final sample size on trunk fat mass. Therefore, after accounting for a 10%

dropout rate and a possible imbalance of up to 40:60 in the two strata, the required sample size was 268.

Recruitment to TRYMS was slower than predicted, so the duration of the trial was extended by 6 months, and a design modification was made to combine the two strata and investigate the influence of baseline serum testosterone in an exploratory analysis. The decision to modify was made without reference to any outcome data. To detect a 1.7-kg difference in trunk fat mass, assuming a dropout rate of 10%, the revised minimum sample size was determined to be 112. A corresponding sample size of between 58 and 144 was required to detect a moderate effect size of 0.5–0.8 in SF36-PF, assuming a dropout rate of 10%.

A statistical analysis plan was agreed on by the trial management group prior to database lock. Analyses were performed using SAS version 9.4 (SAS Institute, Cary, NC, USA).

Baseline variables were reported with descriptive statistics, and no significance tests were conducted.

The primary analysis was conducted on the intention-to-treat (ITT) population, and missing data were imputed using multiple imputation by chained equations. Characteristics of patients with missing data were compared with those without missing data using descriptive summary statistics, and there were no reasons to assume that the missing-at-random assumption was invalid, apart from the DISF-SRII data for which the missing data mechanism could not be determined and a formal analysis was not conducted.

Differences between treatment arms for trunk fat mass at 26 weeks were compared using linear regression adjusting for the clinical minimisation factors, baseline trunk fat mass, and treatment as fixed effects and centre as a random effect. The treatment effect estimate, corresponding 95% CI, and $p$-value are presented.

The SF36-PF data were strongly negatively skewed (overall median 95.0, range 0.0–100.0), and an amendment was made to the original statistical analysis plan so that the SF36 data were dichotomised into perfect SF36-PF (score 100.0) versus nonperfect SF36-PF. Differences between treatment arms for SF36-PF at 13 and 26 weeks were compared using multilevel logistic repeated measures modelling, accounting for data at 13 and 26 weeks and adjusting for the clinical minimisation factors, baseline physical functioning score, time, treatment, and treatment–time interaction as fixed effects. Randomising site, participant, and participant–time interaction were fitted as random effects. Because the study was designed and powered to test the difference in SF36-PF score, not the proportion of perfect SF36-PF score, this analysis should be considered an exploratory end point rather than a primary end point as originally intended.

Differences between treatment arms for BMI, waist circumference, and other QoL scores were compared using multilevel repeated measures modelling with an unstructured covariance matrix, accounting for data at 13 and 26 weeks and adjusting for the clinical minimisation factors, baseline measure of the endpoint, time, treatment, and treatment–time interaction as fixed effects. Randomising site, participant, and participant–time interaction were fitted as random effects.

Differences between treatment arms for fasting insulin:glucose ratio, fasting lipids, lean body mass, whole-body fat mass, and bone density were compared using linear regression adjusting for the clinical minimisation factors, baseline measure of the end point, and treatment as fixed effects, and centre was included as a random effect. Fasting insulin:glucose ratio and triglycerides were analysed on the natural logarithm scale.

Exploratory analyses were conducted to explore treatment effects on trunk fat mass at 26 weeks within subgroups defined for the following baseline characteristics: trunk fat mass (≤13.3, 13.3–17.8 kg, >17.8 kg), serum testosterone (7.0–9.9, 10.0–12.0 nmol/l), age (25–37, 38–50 years), cancer diagnosis (testicular or leukaemia/lymphoma), and LH (low/normal/high

based on the central laboratory reference range). Likelihood ratio tests were used to determine whether the interactions of these characteristics with treatment were statistically significant.

A data monitoring committee with access to unblinded data oversaw the safety of the study. A trial steering committee had oversight of trial progress and approved the protocol changes.

## Results

A total of 136 men aged 25–50 years were recruited between 17 July 2012 and 17 February 2015 (Consolidated Standards of Reporting Trials [CONSORT] diagram Fig 1). The common reasons for ineligibility for registration were age, BMI, time since cancer treatment, or safety exclusion criteria. The commonest reason for ineligibility after registration was serum testosterone outside the range for the study. Baseline characteristics are presented in Table 2. The

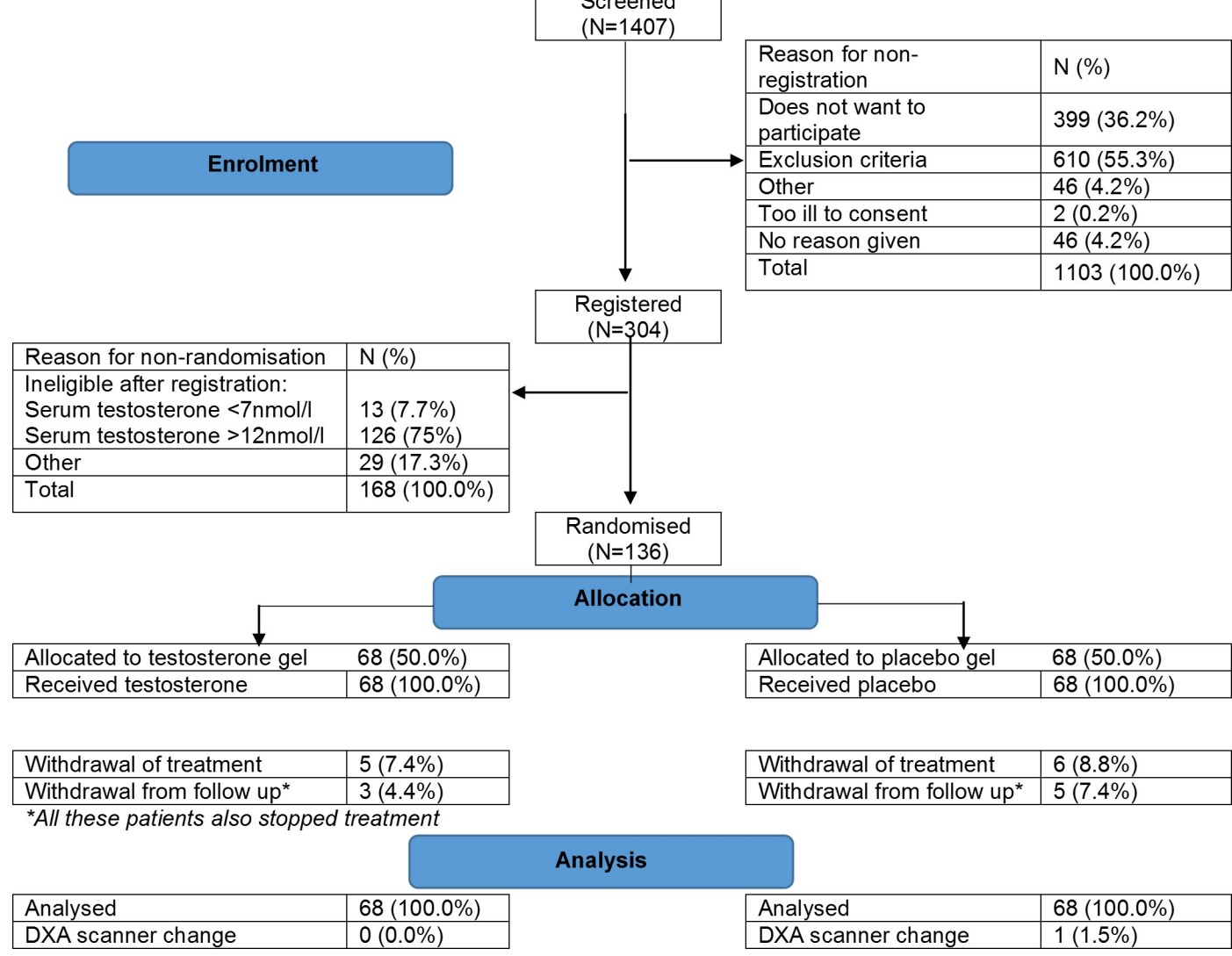

**Fig 1. CONSORT diagram.** CONSORT, Consolidated Standards of Reporting Trials; DXA, dual-energy X-ray absorptiometry.

**Table 2.** Baseline characteristics.

| Characteristic | | Placebo *n* = 68 | Testosterone gel *n* = 68 |
|---|---|---|---|
| Age, years | 25–37 | 29 (42.6%) | 29 (42.6%) |
| | 38–50 | 39 (57.4%) | 39 (57.4%) |
| Height, cm | | 180.0 (5.7) | 181.2 (6.9) |
| BMI, kg/m$^2$ | | 28.1 (3.1) | 27.6 (3.1) |
| Eligibility testosterone, nmol/l | 7.0–9.9 | 41 (60.3%) | 40 (58.8%) |
| | 10.0–12.0 | 27 (39.7%) | 28 (41.2%) |
| Cancer diagnosis | Testicular | 60 (88.2%) | 60 (88.2%) |
| | Lymphoma | 7 (10.3%) | 6 (8.8%) |
| | Leukaemia | 1 (1.5%) | 2 (2.9%) |
| Time from end of cancer treatment, months | 12–30 | 31 (45.6%) | 27 (39.7%) |
| | 31–0 | 18 (26.5%) | 17 (25.0%) |
| | 61+ | 19 (27.9%) | 24 (35.3%) |
| Cancer treatment (not mutually exclusive, 1 participant missing data) | Surgery | 57 (83.8%) | 57 (83.8%) |
| | Chemotherapy | 47 (69.1%) | 48 (70.6%) |
| | Radiotherapy | 8 (11.8%) | 10 (14.7%) |
| Trunk fat mass, kg (mean) | | 15.7 (4.7) | 15.7 (5.3) |
| Trunk fat mass, kg (*N* per tertile) | ≤13.3 | 22 (32.4%) | 24 (35.3%) |
| | 13.3–17.8 | 26 (38.2%) | 19 (27.9%) |
| | >17.8 | 20 (29.4%) | 25 (36.8%) |
| SF36 Physical Functioning score (2 missing) | | 91.9 (14.6) | 81.8 (26.2) |
| Rosenberg Self-Esteem (5 missing) | | 20.9 (5.3) | 19.9 (6.5) |
| DISF-SRII sexual functioning (29 missing) | | 79.3 (25.3) | 74.6 (27.8) |
| FACIT Fatigue (2 missing) | | 35.4 (11.7) | 35.2 (12.1) |
| LH (8 missing) | Normal (1.7–8.6 IU/l) | 44 (64.7%) | 44 (64.7%) |
| | High (>8.6 IU/l) | 22 (32.4%) | 18 (26.5%) |

Continuous variables given as mean (SD), categorical given as N (%).

Abbreviations: BMI, body mass index; DISF-SRII, Derogatis Interview for Sexual Functioning–Self Report II; FACIT, Functional Assessment of Chronic Illness Therapy; LH, luteinising hormone

groups were well balanced for age, height, BMI, eligibility testosterone, type of cancer, and time from cancer treatment. Mean baseline trunk fat mass was similar in the testosterone and placebo groups. The mean baseline SF36-PF score was lower in the testosterone group than the placebo group, and this was accounted for in the analysis.

Eleven (8.1%) participants withdrew from trial treatment before completion: 6 in the placebo group and 5 in the testosterone group. Four participants were unblinded to their treatment allocation: 2 in the testosterone group and 2 in the placebo group. One (in the testosterone group) was due to high serum testosterone at the 2-week titration. Three were required to inform future clinical care. In the placebo arm, 42 participants (62%) had better than 60% adherence, 5 (7%) had less than 60% adherence, and adherence could not be verified in 21 (31%). In the testosterone arm, 43 participants (63%) had better than 60% adherence, 1 (1%) had less than 60% adherence, and adherence could not be verified in 24 (35%).

Median (interquartile range) serum testosterone at 26 weeks was 11.9 (9.9–13.1) nmol/l in the placebo group and 29.3 (15.5 to 43.1) nmol/l in the Tostran group. Median change in serum testosterone from baseline to 26 weeks was −0.4 nmol/l in the placebo group and 16.1 nmol/l in the Tostran group.

## Primary end points

At 26 weeks, testosterone treatment was associated with a statistically significant decrease in trunk fat mass (Fig 2). The treatment effect was −0.9 kg (95% CI −1.6 to −0.3, $p = 0.0073$). In the treatment group, 25% of men had a decrease in trunk fat mass of more than 1.5 kg.

Tostran treatment was not associated with odds of perfect SF36-PF after 26 weeks of treatment (odds ratio [OR] = 0.77, 95% CI 0.30–1.97, $p = 0.5803$).

## Secondary end points

Testosterone treatment was associated with decreased whole-body fat mass (−1.8 kg, 95% CI −2.9 to −0.7, $p = 0.0016$) and increased whole-body lean mass (1.5 kg, 95% CI 0.9–2.1, $p < 0.001$) (Fig 2, Table 3). Treatment was not associated with change in any of the other secondary outcomes: fasting insulin:glucose ratio, fasting lipids, bone density, BMI, waist circumference, SF36, FACIT, or RSE. DISF-SRII sexual functioning results could not be formally analysed because more than 30% of data were missing, and the missing data mechanism could not be assumed to be missing at random due to the sensitive nature of the questions.

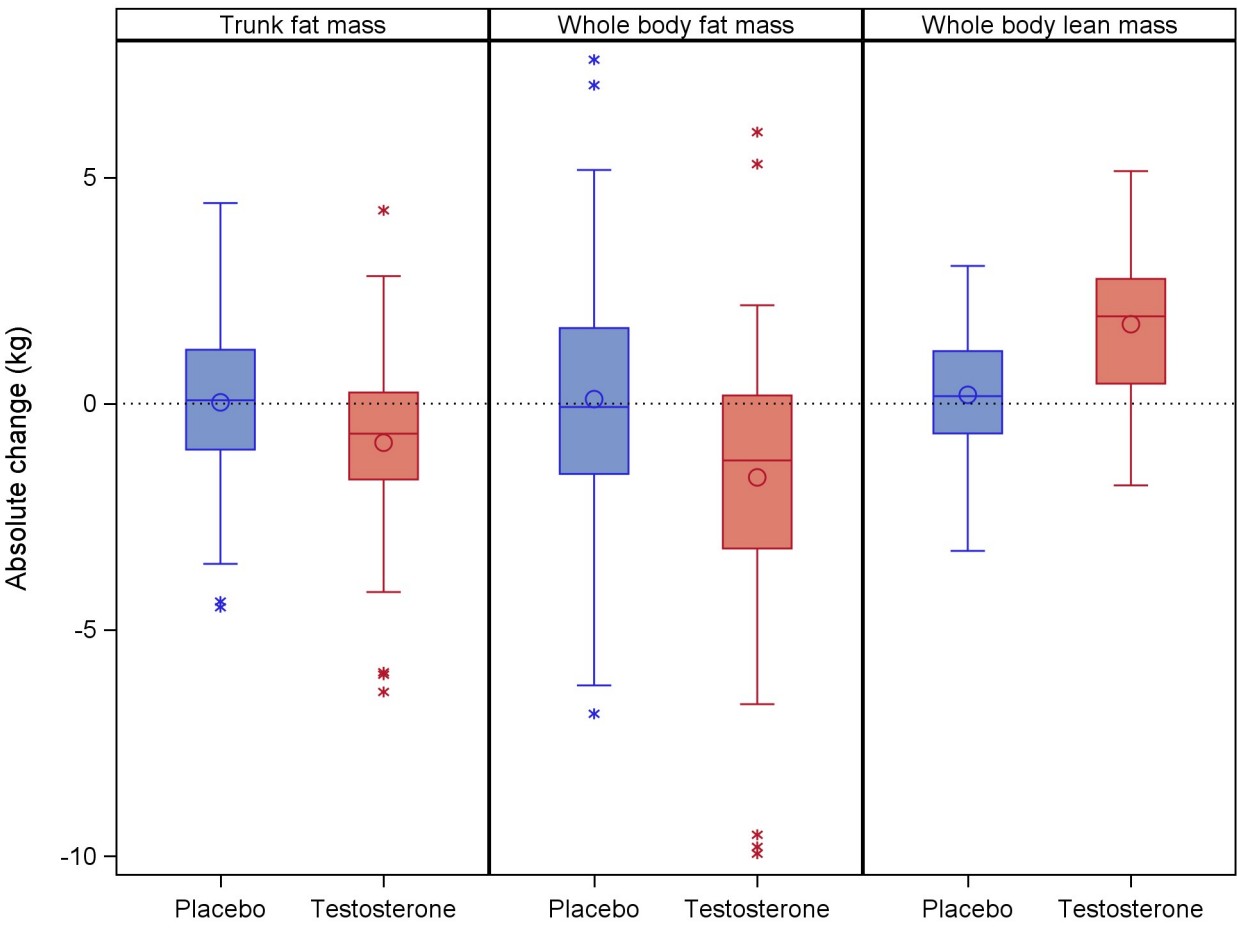

**Fig 2. Change in trunk fat mass, whole-body fat mass, and whole-body lean mass at 26 weeks in placebo and Tostran groups.** Boxes represent the IQR, circles indicate the mean, lines inside the box indicate the median, whiskers represent the range excluding outliers, and asterisks indicate outliers. IQR, interquartile range.

**Table 3. Treatment effect for primary and secondary outcomes at 26 weeks.**

| End point | Treatment effect estimate (95% CI) | p-Value | Analysis method |
|---|---|---|---|
| **Primary** | | | |
| Trunk fat mass, kg | −0.9 (−1.6 to −0.3) | 0.0073 | Linear regression |
| SF36 Perfect Physical Functioning | 0.77 (−0.30 to 1.97) | 0.14 | Repeated measures logistic regression |
| **Secondary** | | | |
| Whole-body fat mass, kg | −1.8 (−2.9 to −0.7) | 0.0016 | Linear regression |
| Lean body mass, kg | 1.5 (0.9–2.1) | <0.001 | |
| Bone density, g/cm$^2$ | 0.00 (−0.01 to 0.01) | 0.42 | |
| Ln(fasting insulin:glucose ratio) | −0.06 (−0.30 to 0.19) | 0.65 | |
| High-density lipoprotein, mmol/l | 0.01 (−0.17 to 0.18) | 0.93 | |
| Low-density lipoprotein, mmol/l | −0.05 (−0.27 to 0.17) | 0.68 | |
| Total cholesterol, mmol/l | −0.03 (−0.26 to 0.21) | 0.83 | |
| Ln(triglycerides) | −0.00 (−0.15 to 0.15) | 0.99 | |
| BMI, kg/m$^2$ | −0.17 (−0.58 to 0.24) | 0.34 | Repeated measures linear regression |
| Waist circumference, cm | 0.16 (−1.81 to 2.13) | 0.87 | |
| SF36 physical component summary | −0.98 (−2.86 to 0.90) | 0.30 | |
| SF36 mental component summary | 0.16 (−3.49 to 3.82) | 0.93 | |
| FACIT Fatigue | −0.63 (−3.40 to 2.13) | 0.65 | |
| RSE self-esteem | −0.40 (−1.99 to 1.20) | 0.63 | |

Abbreviations: BMI, body mass index; FACIT, Functional Assessment of Chronic Illness Therapy; Ln, natural log; RSE, Rosenberg Self-Esteem

## Exploratory analyses

Exploratory analyses found that greater decrease in trunk fat mass with testosterone treatment was associated with older age ($p = 0.031$). The decrease in trunk fat mass was greatest in the highest tertile of baseline trunk fat mass (**Fig 3**), but the treatment interaction term across the three baseline tertiles of trunk fat mass was not statistically significant ($p = 0.20$). The effect of testosterone treatment on change in trunk fat mass was not significantly affected by baseline serum testosterone, cancer diagnosis, or baseline LH.

Baseline and 26-week results for all outcome variables are reported in **Table 4**.

## Adverse events

There were no serious adverse events in the treatment group. Adverse events with a Common Terminology Criteria for Adverse Events (CTCAE) grade three or higher are reported in **Table 5**.

## Discussion

We demonstrated that 6 months of testosterone treatment in young adult male cancer survivors with borderline low testosterone levels improved body composition. Testosterone treatment was associated with decreased trunk and whole-body fat mass of 0.9 and 1.8 kg, respectively, and whole-body lean mass increased 1.5 kg. There was no associated change in QoL or metabolic profile.

Our study is the only double-blind placebo-controlled study sufficiently powered to determine benefits for body composition in young adult male cancer survivors. The one previous placebo-controlled study, in men with lymphoma treated with chemotherapy, found no effect of testosterone treatment on fat mass or lean mass [14]. However, the study had few

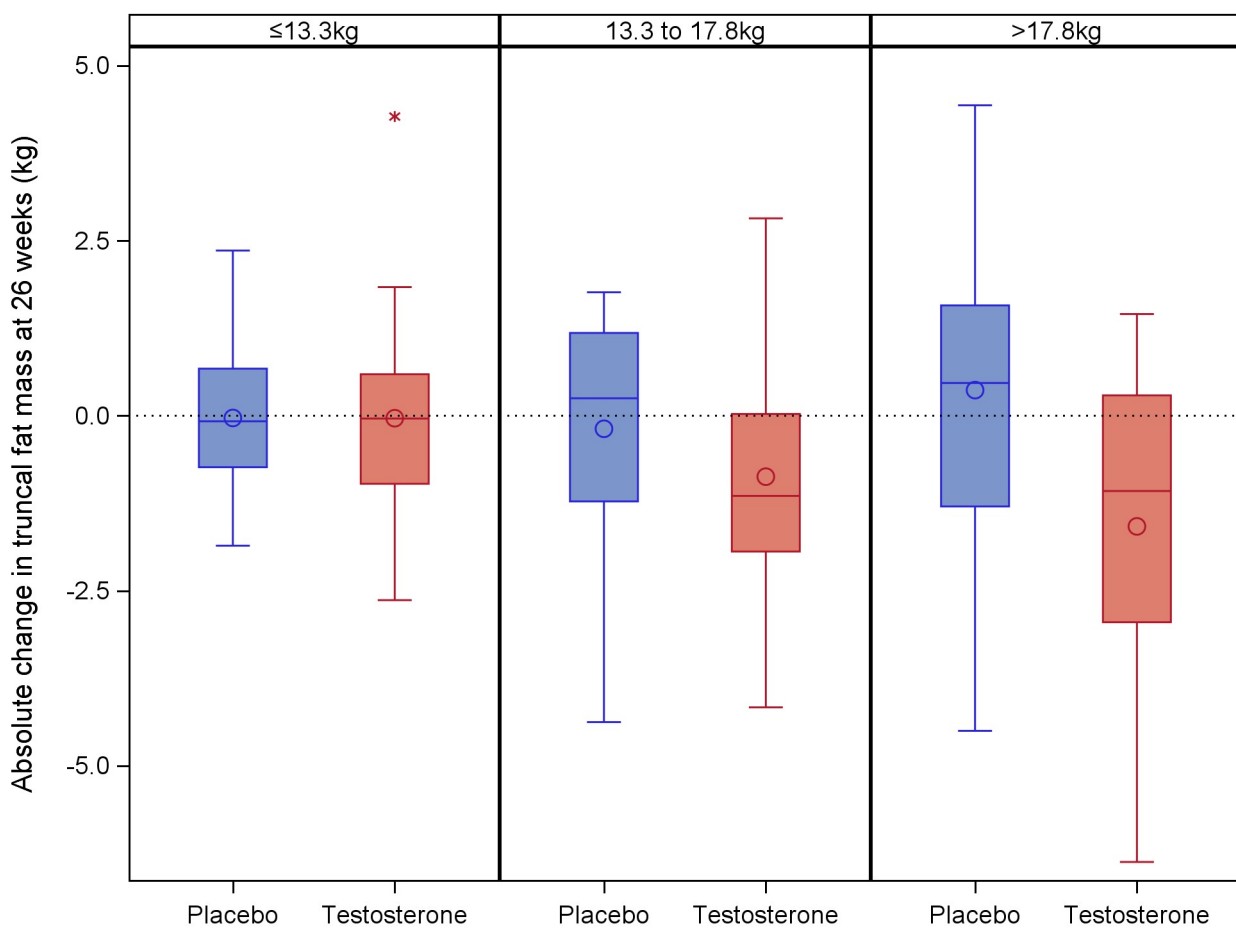

**Fig 3. Change in trunk fat mass at 26 weeks in placebo and testosterone-treated groups by tertile of baseline trunk fat mass.** Boxes represent the IQR, circles indicate the mean, lines inside the box indicate the median, whiskers represent the range excluding outliers, and asterisks indicate outliers. IQR, interquartile range.

participants (total $n$ = 35), they were selected based on raised LH with testosterone in the lower half of the reference range, and there was only a modest increase in serum testosterone from 13.3 to 17.3 nmol/l on treatment. Male cancer survivors have increased fat mass compared with healthy controls [7], but although higher fat mass was associated with lower testosterone levels, it could not be assumed that the lower testosterone was causative for the higher fat mass. Potentially, many factors could influence body composition after cancer treatment, including diet and exercise. Our study does, however, demonstrate that testosterone replacement improves body composition, with an approximately 0.9-kg loss in trunk fat mass; the previously published difference in trunk fat mass between cancer survivors and healthy men was approximately 2.2 kg [7]. These changes occurred with 6 months treatment, and similar changes are reported in hypogonadal men who are not cancer survivors, in which trunk fat mass decreases by 1 kg, whole-body fat mass decreases by 1–3 kg, and lean mass increases by 1–8 kg [13, 19–21]. In older men (age 50–80) with similar baseline testosterone levels and duration of treatment, the effect on body composition was similar to our study [22]. The changes we observed were over a treatment period of 6 months, but based on longer-term studies of testosterone replacement, we could expect these changes to persist with ongoing

**Table 4. Unadjusted absolute values of outcome variables at baseline and 26 weeks.** All given as mean (SD).

| Outcome variable | Placebo *n* = 68 | | Testosterone gel *n* = 68 | |
|---|---|---|---|---|
| | Baseline | 26 weeks | Baseline | 26 weeks |
| Trunk fat mass, kg | 15.6 (4.7) | 15.7 (5.1) | 15.7 (5.3) | 15.0 (5.1) |
| SF36 Physical Functioning | 91.9 (14.6) | 94.8 (10.0) | 81.8 (26.2) | 84.4 (25.7) |
| Whole-body fat mass, kg | 27.2 (7.5) | 27.2 (8.35) | 27.7 (8.2) | 26.1 (8.1) |
| Lean body mass, kg | 59.2 (6.6) | 59.3 (7.1) | 58.2 (7.4) | 60.4 (7.3) |
| Bone mineral density, g/cm$^2$ | 1.28 (0.10) | 1.28 (0.11) | 1.26 (0.11) | 1.27 (0.10) |
| Fasting insulin:glucose ratio | 15.1 (9.5) | 13.7 (8.8) | 14.8 (11.7) | 13.1 (10.2) |
| High-density lipoprotein, mmol/L | 1.52 (1.00) | 1.36 (0.56) | 1.38 (0.73) | 1.28 (0.51) |
| Low-density lipoprotein, mmol/L | 3.37 (0.95) | 3.25 (0.88) | 3.53 (1.16) | 3.34 (1.12) |
| Total cholesterol, mmol/L | 5.25 (1.26) | 5.15 (1.19) | 5.54 (1.23) | 5.36 (1.32) |
| Triglycerides, mmol/l | 1.88 (1.15) | 2.02 (1.48) | 2.13 (1.53) | 2.02 (1.21) |
| BMI, kg/m$^2$ | 28.1 (3.0) | 28.2 (3.3) | 27.6 (3.2) | 27.7 (3.1) |
| Waist circumference, cm | 98.5 (8.7) | 97.6 (8.6) | 98.0 (10.6) | 97.8 (8.7) |
| SF36 physical component summary | 54.2 (7.9) | 54.9 (7.5) | 50.4 (10.0) | 51.0 (10.5) |
| SF36 mental component summary | 41.8 (10.9) | 47.1 (11.1) | 42.9 (12.8) | 47.9 (11.1) |
| FACIT Fatigue | 35.4 (11.7) | 40.1 (10.6) | 35.2 (12.1) | 39.5 (12.0) |
| RSE self-esteem | 20.9 (5.3) | 22.7 (5.8) | 19.9 (6.5) | 21.3 (6.6) |
| DISF-SRII | 79.3 (25.3) | 86.8 (24.6) | 74.6 (27.7) | 85.9 (30.0) |

Abbreviations: BMI, body mass index; DISF-SRII, Derogatis Interview for Sexual Functioning–Self Report II; FACIT, Functional Assessment of Chronic Illness Therapy; RSE, Rosenberg Self-Esteem

treatment [23]. The mean difference in trunk fat mass of 0.9 kg is smaller than the 1.7 kg that we estimated as clinically significant in the power calculation. In the treatment group, 25% of men had a greater than 1.5-kg decrease in fat mass. Subgroup analysis suggested that the change in trunk fat mass with testosterone treatment may be greater in men 38–50 and in those with the greatest baseline trunk fat mass (>17.8 kg), so the effects may be more clinically significant in the men at higher risk of adverse outcomes.

Reference ranges for testosterone in healthy young men vary among laboratories [9]. The Endocrine Society guidelines quote a total testosterone of approximately 10 nmol/l as the lower limit for healthy young men [9]. In this study we stratified male cancer survivors aged

**Table 5. AEs with a CTCAE grade 3 or higher.**

| Number of AEs | AE | CTCAE grade | Related? |
|---|---|---|---|
| **Tostran** | | | |
| 1 | Heartburn/indigestion | Grade 3 | No |
| 1 | Chest pain | Grade 3 | No |
| 1 | Fatigue | Grade 3 | No |
| 1 | Insomnia | Grade 3 | No |
| 1 | Anxiety | Grade 3 | No |
| **Placebo** | | | |
| 1 | Appendicitis | Grade 3 | No |
| 1 | Skin rash | Grade 3 | Yes |
| 1 | Indigestion and bowel disturbance | Grade 3 | No |

Abbreviations: AE, adverse event; CTCAE, Common Terminology Criteria for Adverse Events

25–50 years, according to their screening testosterone taken between 8 and 10 AM, to those with a total testosterone between 7.0 and 9.9 nmol/l and those between 10.0 and 12.0 nmol/l. These levels would be recognised as low or borderline low by most laboratories. The men recruited to this trial correspond approximately to the lowest quartile of serum testosterone in male cancer survivors from our previous observational study [7]. Testosterone levels fall with age and have a circadian rhythm, and normal ranges vary between laboratories. The patients in this study were recruited from multiple centres based on a morning testosterone level taken between 8 and 10 AM. At 26 weeks, the placebo-treated group had mean (interquartile range) total testosterone of 11.9 (9.9–13.1) nmol/l. This probably reflects biological variation and regression towards the mean, but levels remained around the borderline range for young men. The testosterone levels attained in the active treatment arm were at the upper limit of the reference range. The starting dose was according to the license. We included the 2-week dose titration because we felt it was important to achieve robust testosterone replacement and avoid an indeterminate study result due to inadequate testosterone replacement. The titration regime follows clinical practice. It is not possible to know the premorbid testosterone levels of this patient group, and with current testosterone replacement, it is not possible to titrate with perfect accuracy, because there is variable absorption of testosterone through the skin. With these provisos in mind, the aim of our study was to replicate what would be likely to happen in clinical practice. The initial intention was to stratify patients according to their baseline testosterone level, but recruitment was challenging, so we conducted an exploratory analysis of treatment effect according to baseline testosterone. There was no impact of baseline testosterone on response to therapy.

We demonstrated better than 60% adherence in more than 60% of participants, assessed by participant reports and weighing of returned canisters. About 4% of participants were shown to have less than 60% adherence. In the remaining participants, we could not confirm adherence due to nonreturn of canisters. In clinical practice, persistence with topical testosterone treatment is about 35% at 6 months, and the main reason for discontinuation is perceived lack of efficacy [24].

We found no effect of testosterone treatment on QoL assessed with several questionnaires, although because the SF36 data were heavily skewed, which required a change to the statistical analysis plan for this outcome, we cannot definitively confirm that there was no effect on this score. Other studies, including the previous study of men with lymphoma, have reported benefits on QoL measures with testosterone treatment in younger and older hypogonadal men [10, 14, 22, 25]. It may be that the questionnaires we used are not the best tools for this patient population or not sensitive enough to the changes we were trying to detect, and they are not so well validated in younger men [26]. However, in our previous observational study, these questionnaires did identify QoL impairment in cancer survivors compared with healthy controls [8]. Most participants in this study had baseline SF36 scores indicating good QoL, and there may be selection bias in that men with significant fatigue or poor physical or mental QoL are less likely to participate in a 6-month clinical trial than a single-assessment observational study and may have already been treated with testosterone. However, baseline FACIT Fatigue scores were lower (more fatigued) in our current study than the cancer survivors in the previous observational study (35.3 versus 39.6) [8]. On the basis of our current study, we cannot suggest that testosterone treatment improves QoL in this patient group.

Young adult survivors of childhood malignancy have an 8-fold-greater risk of cardiac mortality than the healthy population [27], and cardiovascular morbidity is one of the commonest late effects seen in cancer survivors. Cancer survivors have an increase in all the risk factors for cardiovascular disease including obesity, dyslipidaemia, insulin resistance, and type 2 diabetes [28, 29]. The increased risk particularly affects people who have had higher-intensity

treatments such as haematopoietic stem cell transplantation [30]. If we consider the evidence from population studies, individuals with a BMI between 25 and 29.9 kg/m$^2$ and with one or more risk factors for cardiovascular disease (diabetes, hypertension, dyslipidaemia) have a moderate risk of increased mortality [31]. However, in men treated for testicular cancer, those who had the most intensive treatment were more likely to need prescriptions of antihypertensive and antidiabetes medication, despite having lower BMI than noncancer controls [32], so body composition is not the only factor that affects cardiovascular risk in this patient group. We cannot suggest from this study that testosterone treatment in this patient group improves the metabolic profile. It is important that young male cancer survivors have their cardiovascular risk assessed and individual risk factors treated.

The limitations of our study are a relatively short duration of treatment, only three cancer groups included, and no hard end point data such as cardiovascular events. It is possible that a longer duration of treatment would have had greater effects on secondary end points such as glucose:insulin ratio and lipids. However, the available evidence suggests that any effects on our secondary end points should have occurred by 6 months [33]. Different tissues have different testosterone dose-response relationships [11], but our treatment group achieved serum testosterone near the upper limit of the reference range, so we are confident that they were adequately treated. We were not able to verify adherence in all participants, but verified adherence was similar in the treatment and placebo groups. Although we only included three cancer types, these three make up about 80% of cancers in men of this age, so the results should be widely applicable. We found no effect of cancer type on treatment response, but we recognise that the numbers in the lymphoma and leukaemia groups were smaller than the testicular cancer group. The study did not recruit to the original sample size; however, on the basis that there was no clinical reason to suspect a difference in treatment effects between groups, a redesign to combine serum testosterone groups was made that led to a minimum target of 112 participants. The trial recruited 136 participants, which was close to the maximum number of participants required for the SF36-PF end point on the continuous scale. Although the study was adequately powered for the primary end point of trunk fat mass after the redesign, and minimisation factors were used, there is a possibility in a trial of this size that there was some imbalance between groups, which could have influenced the outcome. We are not advocating that testosterone treatment is promoted for young male cancer survivors; however, this study provides a relevant evidence base for clinicians faced with a young adult male cancer survivor with borderline low testosterone level.

## Conclusions

In young adult male cancer survivors with low and low-normal fasting morning total testosterone 7–12 nmol/l, testosterone treatment is associated with improvement in adverse body composition, and the reduction in trunk fat mass with testosterone treatment is potentially more beneficial in those with an increased trunk fat mass. We suggest that in these patients, testosterone replacement be considered in the context of other interventions to improve body composition.

## Supporting information

**S1 Text. Protocol.**
(DOCX)

## Acknowledgments

We thank Professor Hugh Jones for independent assessment of the dose titrations and Dr Margaret Paggiosi for DXA QA advice. We are grateful to all the participants and local centre

principal investigators: Emilio Porfiri, Danish Mazhar, John Radford, Claire Higham, Ivo Hennig, Daniel Stark, Nicki Panoskaltsis, Steve Knapper, and Ashita Waterston. We would also like to thank the NCRI Clinical Studies Groups who supported the study (Testis, Lymphoma, Haemato-oncology, Psychosocial, and Teenagers and Young Adults), late effects clinic colleagues from around the UK who contributed to the study development, and the members of the trial steering committee and data monitoring committee.

## Author Contributions

**Conceptualization:** Jennifer S. Walsh, Diana M. Greenfield, James Ashton, Robert Huddart, Robert E. Coleman, John A. Snowden, Richard J. Ross.

**Formal analysis:** Isabelle L. Smith, Julia M. Brown.

**Funding acquisition:** Richard J. Ross.

**Investigation:** Jennifer S. Walsh, Emma Best.

**Methodology:** Helen Marshall, Isabelle L. Smith, Emma Best, Julia M. Brown, Richard J. Ross.

**Project administration:** Helen Marshall, Jayne Swain, Richard J. Ross.

**Supervision:** Richard J. Ross.

**Validation:** Richard J. Ross.

**Writing – original draft:** Richard J. Ross.

**Writing – review & editing:** Jennifer S. Walsh, Helen Marshall, Isabelle L. Smith, Diana M. Greenfield, Jayne Swain, Emma Best, James Ashton, Julia M. Brown, Robert Huddart, Robert E. Coleman, John A. Snowden.

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
