## [Decision Letter · Decision Letter 0]

12 Aug 2019

Dear Dr. Ross,

Thank you very much for submitting your manuscript "Testosterone Replacement in Young Male cancer Survivors (TRYMS): A six month double-blind randomised placebo-controlled study" (PMEDICINE-D-19-01858) for consideration at PLOS Medicine. 

[LINK]

In light of these reviews, I am afraid that we will not be able to accept the manuscript for publication in the journal in its current form, but we would like to consider a revised version that addresses the reviewers' and editors' comments. Obviously we cannot make any decision about publication until we have seen the revised manuscript and your response, and we plan to seek re-review by one or more of the reviewers. 

We expect to receive your revised manuscript by Sep 02 2019 11:59PM. Please email us (plosmedicine@plos.org) if you have any questions or concerns.

We look forward to receiving your revised manuscript. 

Sincerely,

Clare Stone, PhD

Managing Editor 

PLOS Medicine

plosmedicine.org

Abstract – Please provide demographic details of participants; the final sentence of the ‘Methods and Findings’ section should include a sentence on the studies limitations; please mention country and setting; please use past tense when you describe the findings 

Funding -I don't think you need to mention the funders in the text

- Please provide an Author Summary – in the main text immediately following the abstract. This should be a non-technical, lay summary. See style for guidelines. 

-Please move the CONSORT flowchart should be in the main paper – usually this would be Table 1 and please reorder others and any shout outs in the text. 

Please provide a table of adverse events?

- At the start of the discussion, we ask you to generalize less and just say what you found in your study (past tense)

- why are there two reference lists?

- Please provide a CONSORT checklist?

Please provide an analysis plan?

Page 10 – you mention multiple questionnaires – are all provided as Supp Files or available without paywall in cited articles? If not, please provide. 

Comments from the reviewers:

Reviewer #1: These authors have carried out a detailed analysis of the effects of testosterone replacement in a group of relatively young cancer survivors: the three groups they chose included testicular cancer, lymphoma and leukemia. The study was meticulously organized and performed, in an exemplary randomized double-blind manner. Unfortunately, the outcomes were fairly negative, showing only a minimal change in body fat but few other important questionnaire or metabolic changes. Nevertheless, these are significant novel data in view of the massive increase in cancer survivors and the few large-scale and placebo-controlled studies.

My major comment is that as the results were relatively disappointing in terms of positive effects, it would be helpful if the authors could offer advice as to which patients should in fact be offered treatment. The changes in body fat were very small, but is there is a level of baseline testosterone in which it should be seriously considered? I cannot see in the present manuscript what threshold of testosterone is worthy of active intervention. 

Minor comments;

1. the adherence rates were similar in the two groups, but were rather low; can the authors discuss this further?

2. A couple of sentences on the changes in levels of testosterone are repeated on page 8.

3. The references are given twice, and are not the same. 

4. All units should also be in SI at first mention and thereafter at every point an international audience.

Reviewer #2: This is a useful RCT on the effect of Testosterone Replacement in Young Male cancer Survivors (TRYMS). The randomisation, sample size calculation, statistical methods and analyses, presentation (tables and figures) and interpretation of the results are mostly adequate and of a good standard. However, there are still a few issue needing attention, especially on the change of the study design and its consequences.

1) I can understand the original plan is to study two strata serum testosterone 200-285 and 286-350 separately to have detailed understanding of the effect of the intervention on these two subgroups. But recruitment difficulty forced the investigators to study a combined group with a broader range of serum testosterone of 200 to 350 instead. The samples size calculation for this combined group is still fine. some issues here: a) where was the effect size of 1.7kg from? references? Were effect sizes same for both the 200-285 and 286-350 groups? b) A sample size of 100 (112 including 10% dropouts) of SF36-PF are only good for effect size 0.6 to 0.8 not the moderate 0.5, so basically only good for detecting relatively bigger effect. c) The final sample size of 136 makes it slightly better but feels like this is a RCT failing to recruit enough patients so failed original purposes and then used the combined group as a compromise. However, the pros and cons of the change in study design and its consequences need to be clearly discussed in the discussion.

2) Sample size and analysis of SF36-PF as co-primary endpoint are not adequate. Firstly the sample size is slightly underpowered for SF36-PF; secondly, to change from a continuous outcome (powered) to binary outcome (not powered and a bit odd using 100 vs <100) makes the analysis not powered and not planned. Therefore the results for SF36-PF are inconclusive and need to tone down the claims of the non-significant findings due to all these compromises.

3) One minor point on table 1, there are many missing % signs for the values in the table.

[LINK]

---

## [Decision Letter · Decision Letter 1]

20 Sep 2019

Dear Dr. Ross,

Thank you very much for re-submitting your manuscript "Testosterone Replacement in Young Male cancer Survivors (TRYMS): A six month double-blind randomised placebo-controlled study" (PMEDICINE-D-19-01858R1) for review by PLOS Medicine.

I have discussed the paper with my colleagues and the academic editor and it was also seen again by reviewers. I am pleased to say that provided the remaining editorial and production issues are dealt with we are planning to accept the paper for publication in the journal.

[LINK]

We look forward to receiving the revised manuscript by Sep 27 2019 11:59PM. 

Sincerely,

Clare Stone, PhD

Managing Editor 

PLOS Medicine

plosmedicine.org

Requests from Editors:

Abstract- what is borderline low testosterone, they should provide a range 

Abstract-which cancers were included? 

Author summary mentions increase in muscle mass, which was not mentioned in the abstract. If included as an endpoint, it should be mentioned sooner. 

Page 4- phrases such as "frankly low" or "well below"

Page 4 ref for the normal range of circulating testosterone- seems odd that they say the normal range in most laboratories. Surely it should be standardised?

Page 6- citations to all questionnaires used in the study should be provided earlier in the endpoints section where they are first mentioned

Page 6- please introduce SST samples on first view

(TRYMS) – remove from title as not a common acronym; Also Is this a trial? If so the study descriptor needs to reflect this (a randomised trial)?

Abstract “ specialised oncology hospital setting” – please say which hospitals or at least cities; please move the summary participant information to the start of the methods and findings section in regard to recruitment; note dates of recruitment should be added to abstract; Please remove ISRCTN: 70274195 and the other trial number from the mid part of the abstract and move to the end of the abstract; The Methods and Findings need to be combined into 1 section, per house style (please see formatting instx) and also the sentence on the limitations of the study need to be moved to the end of this combined section; please explain as background why testosterone is reduced for all cancers, rather than just testicular, as one might expect.

Author summary – what does borderline low level mean….borderline for what?; next point “A common question of young male cancer survivors to their physician is: “would I benefit from testosterone treatment?”” Really? Patients ask this? I wouldn’t have thought that most patients would be aware of it or it be the first thing on their mind – please avoid any declamatory language; Remove final point as it’s repetitive with a point made above. 

Questionnaires – please ensure all are submitted as Supp files

Methods – as requested also in abstract – we need details of hospitals and cities.

Did your study have a prospective protocol or analysis plan? Please state this (either way) early in the Methods section.

c) In either case, changes in the analysis—including those made in response to peer review comments—should be identified as such in the Methods section of the paper, with rationale.

Were there any adverse events? You state calls to collect adverse events were at 6 and 19 weeks – then you say collected from enrolment to final phone call. Please clarify

“Body composition was assessed by whole body DXA. Lunar and Hologic machines were used by the study centres.” – please describe these for the non-specialist

“were measured in local laboratories at each site.” – please be more specific

Results – “July 2012 and February 2015” Please be exact with the dates and please also explain why in the trial registration (https://www.isrctn.com/ISRCTN70274195?q=70274195&filters=&sort=&offset=1&totalResults=1&page=1&pageSize=10&searchType=basic-search) the end point has been changed and also you exceed the stated endpoint. Please ensure any changed to the analysis plan are discussed in the method section.

“The groups were well balanced for age, height, BMI, eligibility testosterone, cancer diagnosis and time from cancer treatment” – please explain how the groups are matched for cancer diagnosis – do you mean type of cancer? 

Please ensure all quantifiable data had 95% Cis and p values

“Our study is the only double-blind placebo controlled study…” – to our knowledge..please amend

“SF-36 scores in our current study were generally good at baseline” – please avoid vague language and be specific

Provide a conclusions heading before ‘In conclusion, ….”

Table 2 – add p values

Fig 1 – please provide more details of the reasons for 610 to be excluded

CONSORT checklist – please provide sections and paragraphs instead of pages as these change ion formatting. 

Comments from Reviewers:

Reviewer #1: All queries satisfactorily answered

Reviewer #2: Thanks authors for their great effort to improve the manuscript. All my questions are comprehensively addressed. I am satisfied with the response and the revision. No further issues needing attention.

[LINK]

---

## [Editor Report · Decision Letter 2]

11 Oct 2019

Dear Dr. Ross, 

On behalf of my colleagues and the academic editor, Dr. Sanjay Basu, I am delighted to inform you that your manuscript entitled "Testosterone Replacement in Young Male cancer Survivors: A six month double-blind randomised placebo-controlled trial" (PMEDICINE-D-19-01858R2) has been accepted for publication in PLOS Medicine. 

PRODUCTION PROCESS

PRESS

PROFILE INFORMATION

Thank you again for submitting the manuscript to PLOS Medicine. We look forward to publishing it. 

Best wishes, 

Clare Stone, PhD

Managing Editor 

PLOS Medicine

plosmedicine.org